# The mammalian LINC complex component SUN1 regulates muscle regeneration by modulating drosha activity

**Tsui Han Loo[1], Xiaoqian Ye[1], Ruth Jinfen Chai[1], Mitsuteru Ito[2], Gisèle Bonne[3], Anne C Ferguson-Smith[2], Colin L Stewart[1]\***

[1]Developmental and Regenerative Biology, Institute of Medical Biology, Singapore, Singapore; [2]Department of Genetics, University of Cambridge, Cambridge, United Kingdom; [3]Center of Research in Myology, Institut de Myologie, Sorbonne Universités, UPMC Univ Paris 06, INSERM UMRS 974, CNRS FRE 3617, Paris, France

**Abstract** Here we show that a major muscle specific isoform of the murine LINC complex protein SUN1 is required for efficient muscle regeneration. The nucleoplasmic domain of the isoform specifically binds to and inhibits Drosha, a key component of the microprocessor complex required for miRNA synthesis. Comparison of the miRNA profiles between wildtype and SUN1 null myotubes identified a cluster of miRNAs encoded by a non-translated retrotransposon-like one antisense (*Rtl1as*) transcript that are decreased in the WT myoblasts due to SUN1 inhibition of Drosha. One of these miRNAs miR-127 inhibits the translation of the *Rtl1* sense transcript, that encodes the retrotransposon-like one protein (RTL1), which is also required for muscle regeneration and is expressed in regenerating/dystrophic muscle. The LINC complex may therefore regulate gene expression during muscle regeneration by controlling miRNA processing. This provides new insights into the molecular pathology underlying muscular dystrophies and how the LINC complex may regulate mechanosignaling.
DOI: https://doi.org/10.7554/eLife.49485.001

**\*For correspondence:**
colin.stewart@imb.a-star.edu.sg

**Competing interests:** The authors declare that no competing interests exist.

## Introduction

In recent years the nuclear envelope (NE) and lamina have attracted much interest due to the identification of a significant number of diseases that are associated with mutations in component proteins of the NE and especially the A-type lamins (*Burke and Stewart, 2014*). The NE is comprised of the inner (INM) and outer (ONM) nuclear membranes, traversed by the nuclear pore complexes. Underlying the NE is nuclear lamina. The lamina is a network of intermediate filament proteins comprised of the A-type and B-type lamins that polymerize to form a meshwork of filaments primarily associated with and underlies the INM (*Aebi et al., 1986*; *Turgay et al., 2017*). The nuclear lamina maintains nuclear shape, provides resistance to mechanical stress, organizes chromatin and acts as a scaffold to localize some 80 integral proteins to the nuclear membranes (*Burke and Stewart, 2013*; *Burke and Stewart, 2014*; *Schirmer et al., 2003*). Among the proteins that depend to some extent on the lamins for their localization, are those of the LINC complex (LInkers of Nucleoskeleton and Cytoskeleton), a protein complex that tethers the interphase nucleus to the cytoplasmic cytoskeleton (*Horn, 2014*). In doing so, the LINC complex provides a direct physical connection between the cell membrane/extracellular matrix to the nuclear envelope and nucleoplasm. With this connectivity it has been proposed that this makes the nucleus a potential mechanosensor (*Alam et al., 2016*; *Kirby and Lammerding, 2018*).

In mammalian cells, the LINC complex is comprised of the SUN and KASH domain proteins. The SUNs 1 and 2, (Sad1p, UNC-84) are evolutionarily conserved genes sharing a common C-terminal SUN domain. Both SUN proteins localize to the inner nuclear membrane (INM). Their N-termini protrude into the nucleoplasm underlying the INM, where the N terminal nucleoplasmic region of SUN1 interacts with pre-Lamin A (*Lmna*) and nuclear pore complexes (*Mattioli et al., 2011*; *Liu et al., 2007*). The C-termini of the SUN proteins extend into the perinuclear space between the INM and ONM where they bind to the C-termini of KASH domains of the KASH (Klarsicht, ANC-1, Syne Homology) family of proteins comprising Nesprins (Nesp) 1–4, KASH5 and LRMP (*Sosa et al., 2012*; *Crisp et al., 2006*; *Haque et al., 2006*).

In mammals six KASH proteins have been identified. The bulk of the KASH proteins extend into the perinuclear cytoplasm where they interact directly or indirectly with the different cytoskeletal networks. Nesprins/KASH 1, 2, 4 and 5 interact with the microtubular network, through interactions with the microtubular motor proteins kinesin and dynein. Nesprins/KASH 1 and 2 also interact with the actin microfilament network via calponin homology domains at their N-termini. Nesprin/KASH 3 interacts with the intermediate filament network via plectin. However, it is unclear if LRMP interacts with any of the cytoskeletal networks (*Horn, 2014*).

The LINC complex regulates nuclear positioning within cells, as well as nuclear migration during muscle and neuronal development. (*Malone et al., 1999*; *Starr et al., 2001*). In mice, SUN1 is required for nuclear positioning in myotube formation, during retinal and neuronal development and in the outer hair cells of the cochlea (*Lei et al., 2009*; *Mattioli et al., 2011*; *Yu et al., 2011*; *Zhang et al., 2009*; *Horn et al., 2013b*). During gametogenesis SUN1, together with KASH5, is also required for the attachment of telomeres to the INM that is essential to bouquet formation of the chromosomes during the first meiotic prophase (*Horn et al., 2013a*). SUN1 is also required for Piwi interacting RNAs (piRNAs) synthesis in the germline (*Chi et al., 2009*).

Besides these cellular functions, the lamins, members of the LINC complex and in other NE proteins, such as emerin are of clinical importance in that mutations result in a range of congenital diseases (*Worman, 2012*). Mutations in the *LMNA* gene result in the laminopathies, consisting of two broad classes of disease (*Burke and Stewart, 2014*). One class affects striated muscle resulting in muscle wasting, dystrophies and cardiomyopathy, such as Autosomal Dominant Emery-Dreifuss muscular dystrophy (AD-EDMD). The other class alter white fat distribution (lipodystrophy), craniofacial and skeletal development (mandibuloacral dysplasia), as well as causing Hutchison-Gilford Progeria, a premature ageing disease that is associated with defects in vascular integrity (*Worman et al., 2010*). Mutations and some variants in the genes encoding other NE proteins, including Emerin, Man1, Lap2α, LBR, Torsin and the KASH and SUN domain proteins, particularly SUN1 and Nesprin/KASH1 have all been associated with a variety of congenital musculoskeletal diseases. The majority of the mutations affect the various types of muscle, including skeletal, cardiac and smooth, suggesting the existence of an integrated network of proteins centred on the nuclear envelope/lamina that are important for muscle homeostasis (*Meinke et al., 2014*; *Li et al., 2014*; *Puckelwartz et al., 2009*; *Puckelwartz et al., 2010*; *Zhou et al., 2017*; *Baumann et al., 2017*; *Chen et al., 2012*).

Given the increasingly recognized importance of the LINC complex in cellular functions and in disease, surprisingly little is known about which proteins/factors, apart from pre-laminA, and nuclear pore complex components interact with the nucleoplasmic domains of the SUN proteins. Since variants in the SUN proteins have been associated with muscular dystrophies (*Meinke et al., 2014*), we sought to identify what other nuclear factors interact with the nucleoplasmic domain of SUN1 in skeletal muscle. Here we show that specific SUN1 isoforms are selectively expressed in human and murine skeletal muscle and that isoform expression changes with muscle differentiation. In vivo, adult mice lacking SUN1 show retarded muscle regeneration. In myoblasts undergoing myotube formation, the nucleoplasmic domain of the predominant SUN1 isoform uniquely binds to the RNase III enzyme Drosha, that initiates microRNA (miRNA) biogenesis (*Roberts, 2015*). Drosha and Pasha (DGCR8) are the core proteins of the Microprocessor complex which regulates miRNA biogenesis in the nucleus (*Han et al., 2004*). They are present as a large molecular weight complex that includes additional accessory proteins in the complex, and increasingly these accessory proteins are being found to regulate the expression and maturation of specific miRNA precursors in a cell specific and developmental context (*Creugny et al., 2018*).

In differentiated myotubes, loss of SUN1 alters the expression levels of a range of miRNAs, including a miRNA cluster derived from the maternally expressed antisense retrotransposon-like 1,

*Rtl1as,* a non-coding RNA transcript. *Rtl1as* is the complementary antisense transcript to the paternally expressed imprinted retrotransposon-like one gene *Rtl1* encoding a protein of unknown function. Previous results revealed that over-expression of *Rtl1,* is associated with muscle hypertrophy, as well as placental growth defects and hepatocarcinoma (*Byrne et al., 2010*; *Ito et al., 2015*; *Riordan et al., 2013*; *Sekita et al., 2008*). Here we show that both *Sun1,* and *Rtl1,* are required for efficient muscle regeneration in adult mice and that SUN1's interaction with Drosha may regulate RTL1 levels by controlling the synthesis of a key miRNA modulating RTL1 translation. These findings identify a pathway by which the LINC complex may regulate protein expression necessary for efficient muscle regeneration by acting as a microprocessor regulatory component.

## Results

### Differentially spliced isoforms of Sun1 are expressed in skeletal muscle and muscle cells in vitro

The *Sun1* gene is expressed as tissue specific differentially spliced isoforms derived by alternate splicing of the 5' exons. These are translated into different SUN1 nucleoplasmic isoforms each with a conserved perinuclear (C-terminal) Sun domain (*Göb et al., 2011*; *Liu et al., 2007*). We analysed *Sun1* cDNA sequences from eight murine tissues (*Figure 1A*). Smaller splice variants were abundant in the CNS (brain), heart, skeletal muscle, and, to a lesser extent, testis. Larger variants were abundant in the kidney, liver, lung, and spleen. We focused on which *Sun1* isoforms were expressed in adult skeletal muscle and found their expression changes during myogenesis. *Sun1* cDNAs from skeletal muscle were sequenced, revealing that smaller transcripts are generated by alternate splicing between exons 7 to 9 (*Figure 1B*). *Sun1* exon splicing was evident during the in vitro differentiation of C2C12 myoblasts into myotubes (*Figure 1C*). In proliferating C2C12 myoblasts, the full length *Sun1* was the most abundant isoform (in 67% of the clones sequenced). This isoform is then replaced by the smaller *Sun1* splice variants, Δ7 and Δ7, Δ9 (in which exons 7 and 9 were deleted with exon eight being retained) and Δ7–9, (in which all 3 exons 7–9 were deleted) as the myoblasts fused to form myotubes.

To determine if these isoforms were conserved between humans and mice, we cloned human SUN1 sequences from human fetal- and adult muscle cDNAs obtained from Clontech. We identified the human SUN1 Δ6 as representing the major isoform expressed in human muscle (data not shown). By aligning the mouse SUN1 and human SUN1 Δ6 amino acid residues, we found many highly conserved residues, except those encoded by exons 7, 8 and 9, in the mouse *Sun1* (*Figure 1—figure supplement 1*). Human SUN1 Δ6 is therefore equivalent to the murine Sun1 Δ7–9 and Sun1 Δ7, Δ9 confirming sequence conservation between the muscle isoforms in both species.

An antibody was raised to the peptide sequence at the junction of exons 6–8 to specifically recognise mouse Sun1 Δ7 and Sun1 Δ7,Δ9 isoforms as previously described (*Calvi et al., 2015*), hereafter referred to as Sun1 Δ7 antibody. SUN1 isoforms with a molecular weight of approximately 75 kDa were detected in skeletal muscle extracts by Western blot analysis using this antibody, which were not detected in muscle from *Sun1*$^{-/-}$ mice (*Figure 1D*).

### *Sun1* is required for efficient muscle regeneration

*Sun1* null (*Sun1*$^{-/-}$) mice are normal at birth, with postnatal growth and development being indistinguishable from that of wild-type (WT) littermates, except that both males and females are sterile, deaf, and have reduced Purkinje cell numbers in the cerebellum (*Chi et al., 2009*; *Ding et al., 2007*; *Horn et al., 2013a*; *Wang et al., 2015*). Sequence variants in *Sun1* have been associated with dystrophic muscle, however there is no report of *Sun1* null mice exhibiting muscle defects (*Meinke et al., 2014*) To determine whether *Sun1* has a role in muscle regeneration we autografted/transplanted specific hind-limb muscle groups to determine their regenerative capability with and without *Sun1*. We chose to use the whole muscle engraftment technique, a highly reproducible and well characterized model for regeneration, as surgical removal of a specific muscle completely severs it from its neuronal and vascular connections, resulting in complete and rapid degeneration of the specific muscle group. If the severed muscle is immediately re-grafted into the same site or onto a different recipient muscle, the grafted muscle regenerates through a well-described and defined series of steps in which the rate of regeneration is quantified (*Roberts and McGeachie, 1992*;

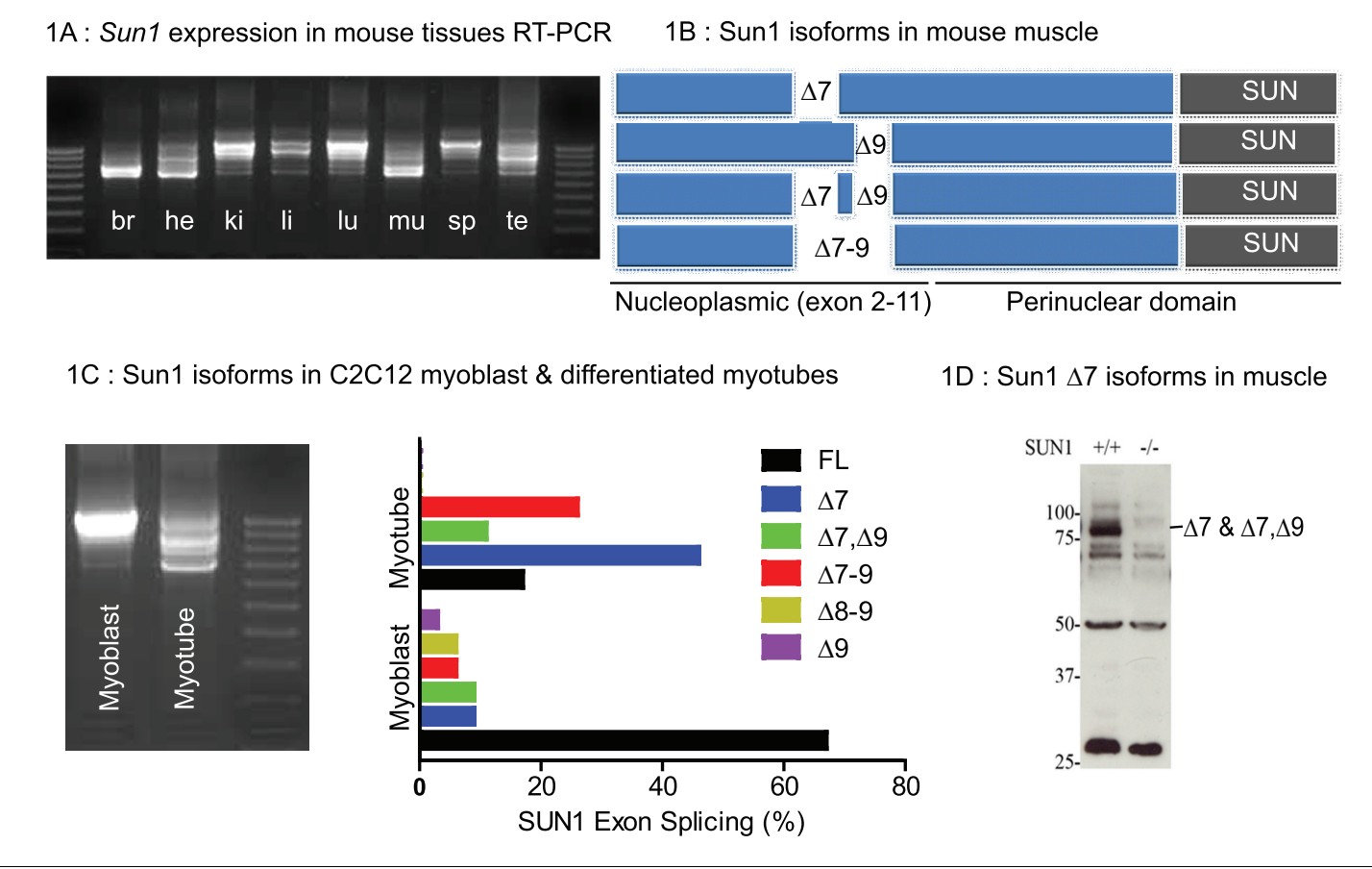

**Figure 1.** Tissue specific Splice Isoforms of Sun1. (A) *Sun1* cDNAs encoding the nucleoplasmic domain were amplified by RT-PCR from murine tissues; brain (br), heart (he), kidney (ki), liver (li), lung (lu), hind-limb muscle (mu), spleen (sp), and testis (te). The outermost lanes are the 100 bp DNA markers. (B) A diagram shows four different *Sun1* cDNAs expressed in muscle due to alternative splicing between exons 7 to 9 (n = 34 clones sequenced). (C) Amplified *Sun1* cDNAs from C2C12 myoblasts and differentiated myotubes showing major differences in expression of the spliced variants. These cDNAs were cloned and sequenced (n = 34 clones) with the *Sun1* isoform percentages being presented in the chart, together with the full-length transcript. (D) Western blot showing the Sun1 Δ7 antibody specifically recognizes ~75 kD SUN1 protein in skeletal muscle lysates and is not detected in *Sun1$^{-/-}$* muscle.

DOI: https://doi.org/10.7554/eLife.49485.002

The following figure supplement is available for figure 1:

**Figure supplement 1.** Amino acid sequence alignment of mouse SUN1 and human SUN1Δ6.

DOI: https://doi.org/10.7554/eLife.49485.003

*Shavlakadze et al., 2010*; *White et al., 2000*). The advantage of this procedure over the more conventional cardiotoxin injection method is that this procedure allows for the detection of both muscle autonomous and host environmental effects on the regeneration process.

We excised the entire Extensor Digitorum Longus (EDL) muscle (from either *Sun1$^{-/-}$* or WT donors, and grafted it onto the Tibialis Anterior (TA) muscle in 2–3 month old mice (*Roberts and McGeachie, 1992*; *Shavlakadze et al., 2010*). The EDL grafts were allowed to regenerate for 9 days and then recovered for analysis. In *Figure 2A*, a cross section of the WT EDL graft with the neighbouring recipient TA muscle is presented. Muscle regeneration progresses from the periphery towards the centre of the graft. During regeneration, inflammatory cells (neutrophils, macrophages) are recruited to the graft, necrotic muscle cells are phagocytosed, the extracellular matrix is reorganized, while activated satellite cells proliferate and fuse to form new myofibers to replace the necrotic ones. In WT donor grafts, at day nine post-engraftment, the graft had regenerated new myofibers with little remaining necrotic muscle at the core (*Figure 2*; images A, B and E). In the magnified image E, examples of regenerated myofibers with centrally positioned nuclei (marked *) and necrotic

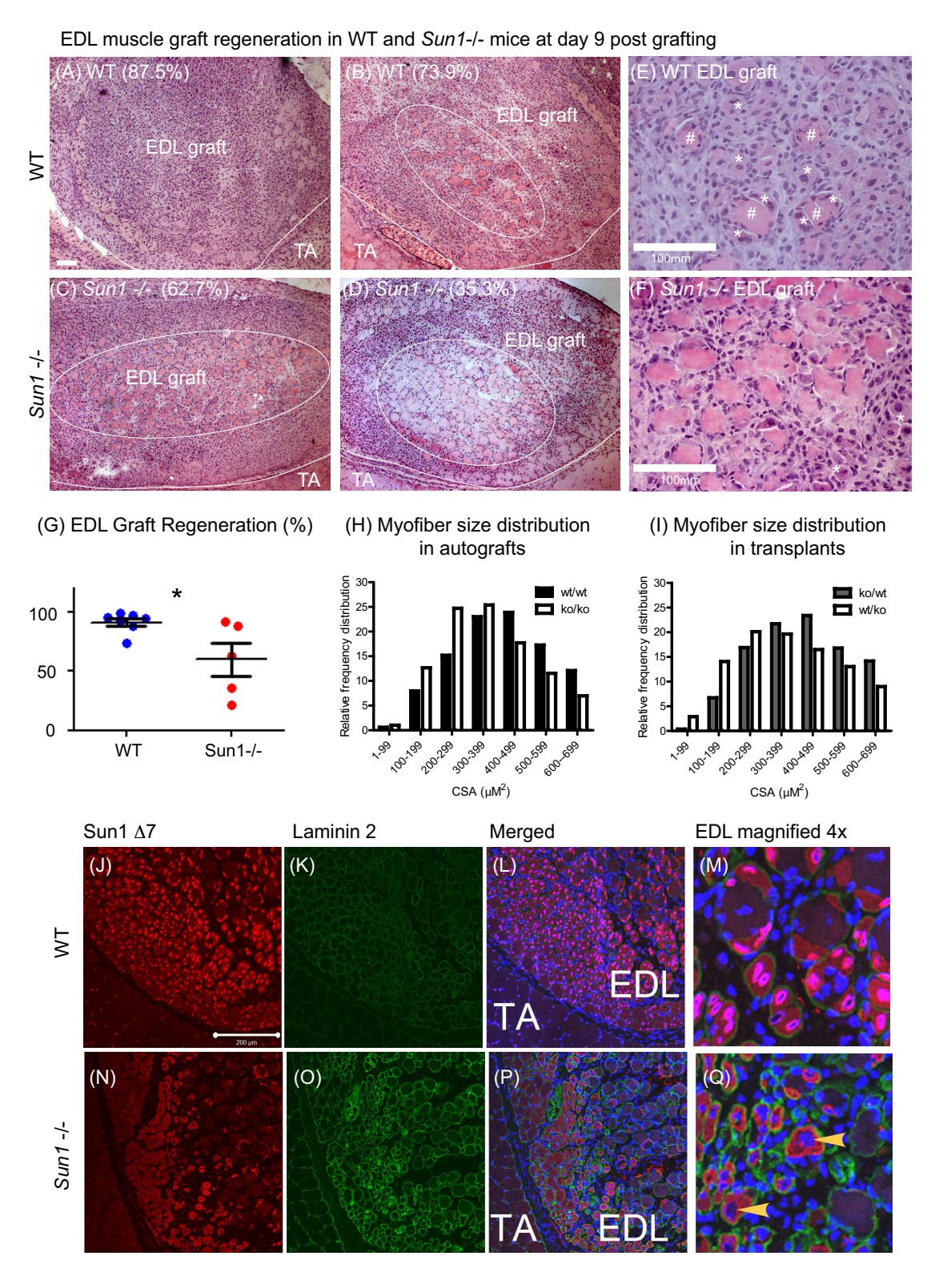

**Figure 2.** Loss of SUN1 retards muscle regeneration. (**A**) Haematoxylin and eosin (**H and E**) stained images of EDL grafts at day nine post engraftment. In the WT grafts (**A and B**) and *Sun1*[−/−] grafts (**C and D**) the central necrotic muscles are demarcated by a white oval, with the boundary of the EDL graft and TA muscle being marked with a white line. Scale bar, 100 µm. (**E**) Magnified view of necrotic muscle fibers (#), with regenerating myofibers (*) clustered around the necrotic muscle and clusters of regenerated muscle fibers (*) in the WT EDL graft. (**F**) Myogenesis had initiated at the

*Figure 2 continued on next page*

*Figure 2 continued*

*Sun1*−/ /− periphery (*), however the core area was occupied with necrotic muscle fibers. **(G)** Comparison of muscle regeneration between WT (n = 7) and *Sun1*−/− (n = 5) mice. The numbers are expressed as a percentage of newly regenerated myofibers over total muscle numbers in the graft, with mean ± SEM and *p<0.05. Percentage of regeneration is included (in parenthesis) in the H and E images. **(H and I)** The cross sectional area (CSA) of the regenerated myofibers in both autografts and reciprocal grafts were quantified and presented in CSA distribution curves. **(J)** SUN1 Δ7 and - Δ7, Δ9 isoforms localize to the NE of regenerating EDL myofibers in vivo. Images (red immunostaining) of (top panel) SUN1 Δ7 at the nuclear envelope (NE) of WT muscle fibers; **(K)** laminin 2 (green) staining defines the boundary of individual myofibers and **(L)** with DAPI staining to mark the nuclei. **(M)** Sun1 Δ7 and - Δ7, Δ9 isoforms (red immunostaining) are expressed in regenerated muscle myofibers NEs (defined by the green laminin staining), with the isoforms being absent in surrounding non muscle cell nuclei (blue). **(Q)** In the *Sun1*−/− grafts the Sun1 Δ7 antibody did not stain the NE of regenerated muscle myofibers (as defined by the green laminin staining and yellow arrowheads), (bottom panel). The red cytoplasmic signal in myofibers is non-specific probably due to cross-reactivity by the polyclonal antibodies. Scale bar, 200 μm.

DOI: https://doi.org/10.7554/eLife.49485.004

The following figure supplements are available for figure 2:

**Figure supplement 1.** Number of regenerated myofibers in reciprocal graft transplants; WT EDL into *Sun1*−/− host (ko/wt) and *Sun1*−/− EDL into WT host (wt/ko) reveals no significant differences.

DOI: https://doi.org/10.7554/eLife.49485.005

**Figure supplement 2.** Sun1 expression in differentiating myoblast cultures.

DOI: https://doi.org/10.7554/eLife.49485.006

ones (marked #) are highlighted. In contrast in the *Sun1*−/− donor grafts (**Figure 2**; images C, D and F), necrotic muscle fibres persisted and occupied more than half the central region of the graft. Although myogenesis had initiated at the graft periphery, the persistence of necrotic muscle fibers revealed that myogenesis was defective (**Figure 2F**). The total numbers of regenerated myofibers and necrotic myofibers were quantified in each graft and expressed as percentage of muscle regeneration (**Figure 2G**) with regeneration being significantly retarded (p<0.05) in the *Sun1*−/− grafts at D9 after transplantation.

The persistent presence of necrotic myofibers in the *Sun1*−/− grafts probably impedes new myofiber synthesis, as previously demonstrated in other muscle grafts (**Shavlakadze et al., 2010**). We performed reciprocal grafting (WT EDL into *Sun1*−/− host 'ko/wt' and *Sun1*−/− EDL into WT host 'wt/ko') to further address the potential role(s) of SUN1 in necrotic muscle clearance and myogenesis. However, there was no difference in the number of regenerated myofibers at day nine post-grafting (**Figure 2—figure supplement 1**). On average, normal regeneration results in the formation of between 400–600 myofibers per graft (**Shavlakadze et al., 2010**). Three WT grafts that had fewer new myofibers, also had necrotic myofibers that were not efficiently cleared in the *Sun1*−/− host (ko/wt). For *Sun1*−/− grafts transplanted into WT hosts (wt/ko), necrotic myofibers in the grafts were cleared by day nine with the formation of new myofibers. **Figure 2H and I** show the myofiber size distribution in these grafts, with *Sun1*−/− myofiber size distribution being smaller than WT myofibers in both the autografts and transplants into a different host. These findings suggest that *Sun1* may have distinct roles in both the host environment and the muscle graft itself during regeneration, although it's loss in the regenerating muscle results in myofibers with a smaller diameter.

SUN1Δ7 isoform expression and localization was analysed in these regenerating muscle fibres both in vivo and in vitro in myoblast cultures established from WT and *Sun1*−/− mice. The SUN1 Δ7 antibody strongly stained the NEs of newly formed WT myofibers within the EDL graft, and to a much lesser extent the peripheral nuclei of the host TA muscle (**Figure 2J,L and M**). Interstitial cells in the WT graft did not express SUN1 Δ7. SUN1 was not detectable in the myofiber nuclei in the *Sun1*−/− grafts (arrowheads) using the SUN1 Δ7 antibody (**Figure 2N,P and Q**). Laminin two staining defined the myofiber boundary and DAPI marked the centrally located nuclei in these newly formed myofibers.

The Sun1 Δ7 antibody localized SUN1 to the LaminB1 positive nuclear envelopes (NE) of WT nuclei in myotubes differentiated in culture (**Figure 2—figure supplement 2B** upper panels), with *Sun1*−/− myofibers not showing any antibody localization to the NEs confirming its specificity (**Figure 2—figure supplement 2B** lower panels). Intriguingly however, in vitro, the loss of SUN1 during myotube differentiation did not overtly impede the differentiation of myoblasts into myotubes (**Figure 2—figure supplement 2C**).

## The *Sun1* Δ7,Δ9 isoform interacts with the microprocessor components Drosha and Pasha (DGCR8)

The N-terminus of SUN1 protrudes into the nucleoplasm where it potentially could interact with nucleoplasmic proteins. Apart from pre-laminA and nuclear pore complexes (*Mattioli et al., 2011*; *Liu et al., 2007*), no other protein has yet been reported to interact with SUN1's nucleoplasmic domain. We used the *Sun1 Δ7,Δ9* isoform as a bait in a yeast two hybrid screen to identify nuclear proteins potentially binding to the nucleoplasmic domain of SUN1 and was performed using the Myriad ProNet system (*Supplementary file 1*). The RNase III protein Drosha was repeatedly isolated from two different libraries in the screen. The SUN1–Drosha protein-protein interaction was validated by both immunofluorescence analysis and co-immunoprecipitation (*Figure 3A and B*). Immunostaining of differentiated myotubes with anti-Drosha antibodies showed enrichment at the NE, together with SUN1 (*Figure 3A* top panel), while Drosha was not enriched at the NE in $Sun1^{-/-}$ myotubes suggesting that Drosha is recruited to the nuclear periphery/NE in the presence of SUN1Δ7,Δ9 isoform. To validate this protein-protein interaction, the Sun1 Δ7 antibody was used to co-precipitate SUN1-Drosha protein complexes from myotube lysates (*Figure 3B*). This interaction was specific, as Drosha did not co-precipitate with the other SUN1 isoform, Δ7–9, and HA epitope antibodies.

FLAG-tagged SUN1 Δ7, Δ9 and GFP-tagged Drosha cDNAs were simultaneously expressed in NIH3T3 cells and showed co-enrichment at the NE by immunofluorescence analysis (*Figure 3C*). FLAG-tagged SUN1 Δ7,Δ9 localised to the NE, while anti-GFP antibodies localized the GFP-tagged Drosha, at both the NE and in the nucleoplasm, with Drosha concentrating at the NE. Drosha expression on its own was present throughout the nucleoplasm (*Figure 3C* middle panel).

As Drosha and the RNA binding protein, Pasha (DGCR8), are part of the microprocessor protein complex (*Denli et al., 2004*), we performed a co-precipitation analysis of Pasha and Drosha with mouse SUN1 Δ7,Δ9 or human SUN1Δ6 (*Figure 3D*). FLAG/HA-Pasha, full length Drosha or the N-terminal Drosha fragment, mouse MYC-SUN1 Δ7,Δ9 or human SUN1Δ6-FLAG/MYC were transiently expressed in HEK293 cells. Anti-FLAG antibodies co-precipitated both full length Drosha-GFP and mouse MYC-SUN1 Δ7, Δ9 proteins with FLAG/HA-Pasha (*Figure 3D* lane 1). Similarly, anti-MYC antibodies co-precipitated FLAG/HA-Pasha, and Drosha with human SUN1 Δ6-FLAG/MYC protein (*Figure 3D* lanes 2 and 3). The diagram shows the N- terminal Drosha segments (truncated fragments identified by Y2H) as the SUN1 binding domain, and is partially overlapping with the Pasha binding domain (*Han et al., 2004*). Together these findings indicate that the nucleoplasmic domain of the SUN1 muscle specific isoform Δ7,Δ9 interacts with the Drosha/Pasha microprocessor complex, enriching Drosha localization at the nuclear periphery/NE (*Figure 3A*).

## *Rtl1as* encoded miRNAs are increased in *Sun1–/–* myotubes

Since the muscle enriched SUN1 isoform binds to Drosha, we then analysed whether the loss of SUN1 alters muscle miRNA profiles. We also investigated whether SUN1's interaction with Drosha would affect the processing of primary miRNAs (pri-miRNAs) to precursor miRNAs (pre-miRNA) (*Lee et al., 2003*; *Tomari and Zamore, 2005*).

The loss of SUN1 in the $Sun1^{-/-}$ myotubes did not alter either Drosha or Lamin A protein levels (*Figure 4A*). We then compared the mature miRNA expression profiles between $Sun1^{-/-}$ and WT myotubes by microarray analysis (performed by Exiqon). Nine miRNAs were identified that differed significantly in their expression levels between WT and $Sun1^{-/-}$ with some showing increased and others decreased levels of expression in the $Sun1^{-/-}$ myotubes (*Figure 4B*). Of these nine miRNAs, we chose to focus on three that significantly increased in the $Sun1^{-/-}$ myotubes, miRNAs, −127–3 p, −434–3 p, −431–3 p, as they are all encoded as a cluster within the *Rtl1* antisense transcript (*Rtl1as*) (*Figure 4C* panel one and lower diagram).

We focused on these miRNAs as the *Rtl1/Rtl1as* region is a genomically imprinted locus that previously has been implicated in regulating skeletal muscle growth and therefore they would be of functional relevance to understanding muscle growth and regeneration (*Davis et al., 2005*). Overexpression of the paternally expressed RTL1 locus (or PEG11), which encodes a neo-functionalized retrotransposon protein of unknown function, results in increased muscle mass (skeletal muscle hypertrophy). However the levels of RTL1 protein are regulated post-transcriptionally by the miRNAs encoded by the complementary maternally expressed *Rtl1as* (PEG11as) transcript that acts as a

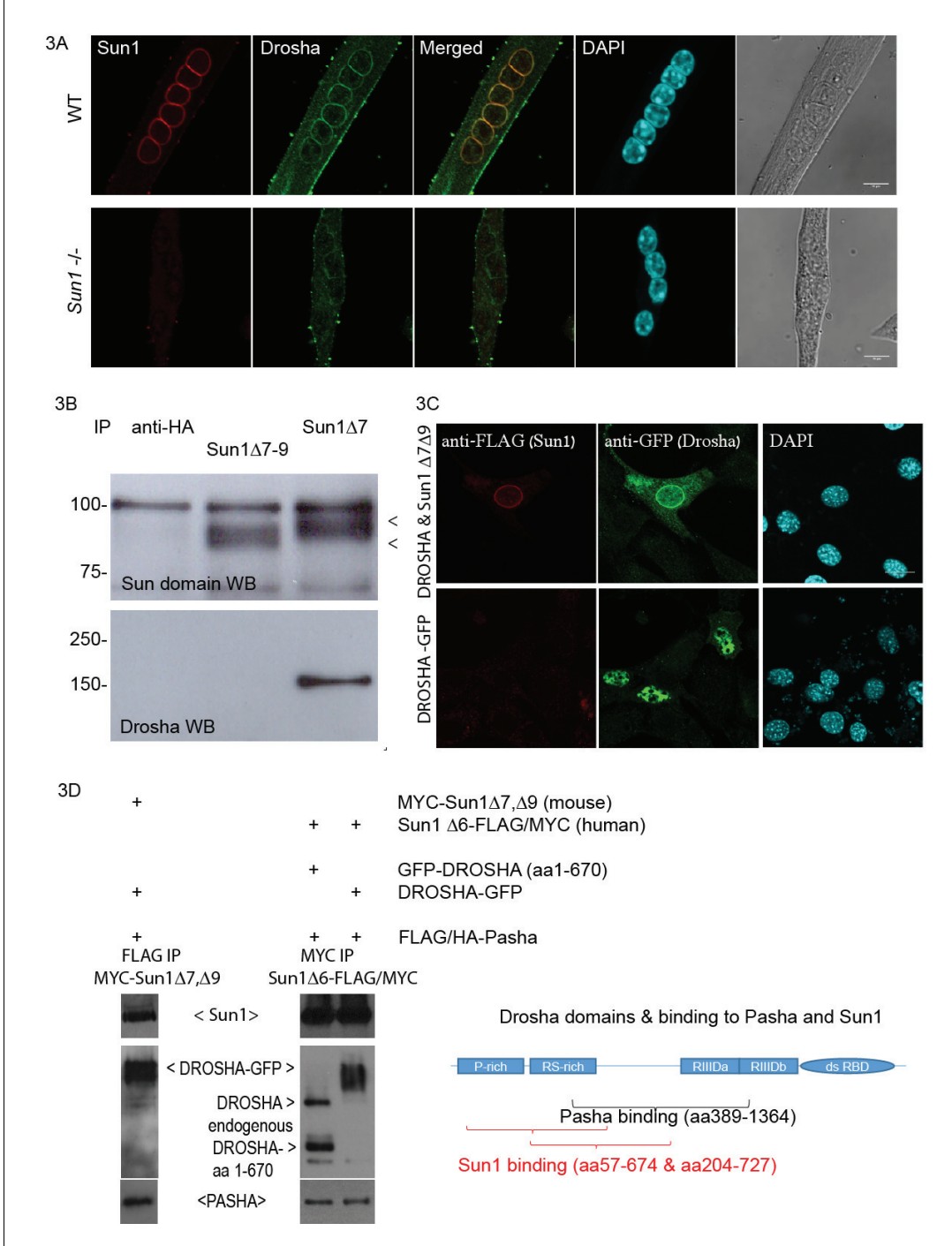

**Figure 3.** Sun1 interacts with Drosha. (**A**) Confocal images of Drosha and SUN1 isoforms (12.10F antibody recognizing all SUN1 isoforms) localized to the NE of WT myotube nuclei, but not in the *Sun1$^{-/-}$* myotubes. (**B**) Immuno-precipitation of SUN1 isoforms from C2C12 myotubes with two SUN1 antibodies (Δ7 and Δ7–9). SUN1 immuno-precipitates were detected with an anti-SUN domain antibody. Drosha was co-precipitated with SUN1 Δ7, but not with the Sun1 Δ7–9 isoform or anti-HA control antibodies. (**C**) Transient transfection of tagged cDNAs of Drosha and SUN1 Δ7 isoform in NIH3T3 cells, reveals enriched GFP tagged Drosha to the NE in nuclei co-expressing FLAG tagged SUN1 Δ7, Δ9 (top panel). Drosha expression in the absence of SUN1 Δ7,Δ9 localized throughout the nucleoplasm (bottom panel). (**D**) Transient transfection of tagged cDNAs in HEK293 cells for co-immuno-precipitation analysis. The Drosha schematic diagram shows the proline rich-, arginine/serine rich domains, the catalytic domains (RIIID) and RNA binding domain. The Pasha- and Sun1 binding domains partially overlap.
DOI: https://doi.org/10.7554/eLife.49485.007

### 4A : DROSHA and LaminA expression levels are unchanged by the loss of Sun1

WT KO

Nucleus

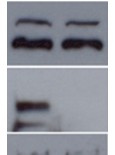

DROSHA

Sun1 Δ7

Lamin A

### 4B : Most significant differences in miRNA expression levels between WT and Sun1-/- muscle

| microRNA | p-value | LMR | | dLMR | fold change |
| | | average WT | average KO | | |
|---|---|---|---|---|---|
| mmu-miR-543 | 1.11E-05 | -0.67 | 0.03 | 0.69 | 1.62 |
| mmu-miR-127 | 2.56E-05 | -0.57 | 0.42 | 0.99 | 1.99 |
| mmu-miR-298 | 6.98E-05 | 0.5 | -0.67 | -1.17 | 0.44 |
| mmu-miR-434-3p | 7.92E-05 | -0.53 | 0.32 | 0.85 | 1.81 |
| mmu-miR-431 | 1.20E-04 | -0.53 | 0.38 | 0.91 | 1.88 |
| mmu-miR-329 | 3.40E-04 | -0.28 | 0.18 | 0.46 | 1.38 |
| mmu-miR-296-3p | 4.60E-04 | 0.28 | -0.66 | -0.94 | 0.52 |
| mmu-miR-145 | 7.39E-04 | -0.16 | 0.15 | 0.31 | 1.24 |
| mmu-miR-411* | 1.00E-03 | -0.38 | 0.4 | 0.78 | 1.72 |

### 4C : *Rtl1as* encoded miRNAs are increased in *Sun1-/-* myotubes

Table 1 (reproduced from the microarray data Fig 4B)

| Host gene | Associated MicroRNAs | Fold change | p-value |
|---|---|---|---|
| *Rtl1as* | *mmu-miR-127* | 1.99 | 2.55554E-05 |
| *Rtl1as* | *mmu-miR-434-3p* | 1.81 | 7.91875E-05 |
| *Rtl1as* | *mmu-miR-431* | 1.88 | 0.000120324 |
| *Rtl1as* | *mmu-miR-433* | 1.55 | 0.002815426 |

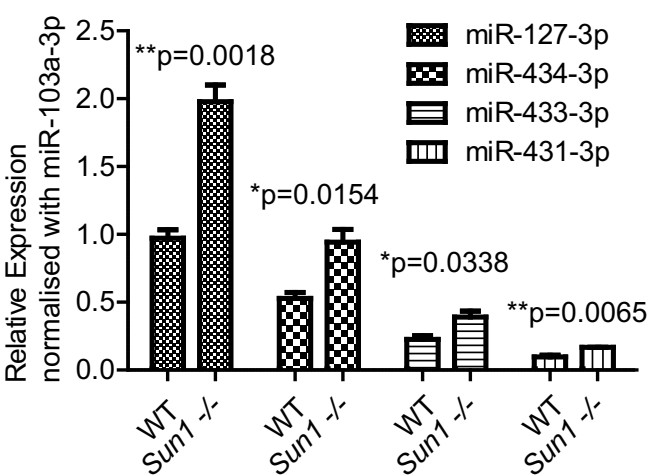

Validation by qPCR

Schematic of *Rtl1as* and the encoded miRNAs, the blue highlighted ones are conserved in humans

| miR-431 | miR-433 | miR-127 | miR-434 | miR-136 |
|---|---|---|---|---|
| 279-369 | 1547-1670 | 2678-2747 | 4338-4431 | 5159-5220 |

**Figure 4.** Effect of Sun1 loss on DROSHA expression and on the regenerating myofibre miRNA profile. (**A**) Equal Drosha and Lamin A protein levels were present in the nuclear fractions from *Sun1[−/−]* and WT myotube cultures. WT and *Sun1[−/−]* primary myoblasts were differentiated into myotubes and then processed into cytoplasmic and nuclear fractions. Sun1 Δ7 antibody was the control for the WT lysate, Lamin A antibody was the positive control for the nuclear fraction. (**B**) Table lists the miRNAs that show significant fold differences in expression levels between WT and *Sun1[−/−]* myotubes. The p-values are based on the t-test. The top five miRNAs pass the Bonferroni correction for multiple testing (0.000161). Average expression (LMR) are shown for both groups, including the difference (dLMR) and the conversion into fold change. (**C**) The four miRNAs are all encoded by the *Rtl1as* transcript (see underlying schematic of the *Rtl1as* transcript and the position of the miRNAs). These miRNAs were expressed at significantly higher levels in *Sun1[−/−]* myotube cultures compared to WT cultures (4C Right panel). The relative abundance of each miRNA is also shown – (miR-127–3p>miR-434–3 p>miR-433–3p>miR431-3 p) and was determined by the CT values (qRT-PCR).

DOI: https://doi.org/10.7554/eLife.49485.008

The following figure supplement is available for figure 4:

*Figure 4 continued*

**Figure supplement 1.** RTL1 protein was undetectable in WT primary myoblasts derived from hindlimb muscle, *Rtl1* null derived primary myoblasts were a negative control (Left panel).

DOI: https://doi.org/10.7554/eLife.49485.009

repressor for RTL1. The four *Rtl1as* encoded miRNAs expression levels are of significance, as these, especially miR-127–3 p, suppress *Rtl1* translation by RISC-mediated cleavage of the *Rtl1* mRNA (*Hagan et al., 2009*; *Ito et al., 2015*; *Lin et al., 2003*; *Sekita et al., 2008*). Mutations in the *Dlk1-Dio3* region, within which the *Rtl1* locus is localized, appear to disrupt the expression of *Rtl1s*, resulting in increased RTL1 protein levels. These in turn lead to muscle hypertrophy in both Callipyge sheep and in transgenic mice engineered to overexpress *Rtl1* (*Davis et al., 2004*; *Xu et al., 2015*).

Previous studies showed that *Rtl1as* RNA transcripts are processed into these miRNAs, as well as a 4th member of the cluster miR-433, by the microprocessor complex (*Davis et al., 2005*). We validated the miRNA levels by quantitative PCR analysis of myotube cDNAs and found that the *Rtl1as* miRNAs were all increased, albeit to differing extents in the $Sun1^{-/-}$ myotubes, with miR-127–3 p being processed at the highest levels and miR-431–3 p at the lowest (*Figure 4C*).

RTL1 protein levels in primary myoblast cultures were undetectable, in comparison to the expression levels in fetal leg muscle (*Figure 4—figure supplement 1*). This therefore precludes any analysis of RTL1 functions in primary myoblast/myotube cultures.

## Sun1 regulates *Rtl1as* processing and *Rtl1* expression during muscle regeneration

Adult muscle autografts were then analysed to determine whether SUN1 loss affected *Rtl1as* and *Rtl1* expression during regeneration. We examined the Drosha-mediated cleavage products of *Rtl1as* in these muscle grafts, specifically those of miR-127, miR-433 and miR-431 sequences by the RLM-RACE protocol (*Figure 5*). The miR-127, miR-433 and miR-431 RNAs were ligated to a common RNA adaptor, reverse transcribed into cDNA and cloned. DNA sequencing revealed that Drosha cleaved the 3′ strand of these three pri-miRNAs (marked by the blue asterisk), as previously reported (*Davis et al., 2005*). Unexpectedly, a larger cDNA of miR-127 (marked by the red asterisk) showed that Drosha cleaved the 5′ strand of miR-127 but the 3′ strand remained intact (summarized in *Figure 5E*). This aberrant pri-miR-127 product was present at 2-fold higher levels in WT grafts compared to $Sun1^{-/-}$ grafts (*Figure 5B*). Under normal circumstances, Drosha processing would produce the 2 nucleotide 3′ overhang of a pre-miRNA necessary for interaction with exportin-5 to exit the nucleus (*Wu et al., 2018*). Misprocessing of pri-miR-127 would lead to less mature miR-127, as seen in relatively lower levels of pre-miR-127 in WT when compared to $Sun1^{-/-}$ muscle (*Figure 5C*). We then addressed if SUN1 interaction with Drosha directly affected the processing of pri-miR-127 in vitro. Drosha, PASHA and SUN1 proteins were ectopically expressed and then immunoprecipitated from HEK293 cells. These proteins were incubated with pri-miR-127 RNA transcribed in vitro. As expected, Drosha RNase activity cleaved the pri-miR-127 to pre-miR-127 (*Figure 5D* lane 3). However, in the presence of SUN1 protein, Drosha activity was impaired in that pri-miR-127 was processed largely into intermediate miR-127 RNA forms (*Figure 5D* lane 4).

As anticipated, *Rtl1* transcript levels were significantly increased in the post-natal muscle (D24) from *miR-127* null mice compared to their WT littermates (*Figure 5—figure supplement 1*). These results therefore suggest that the interaction of SUN1 with the microprocessor complex leads to increased levels of aberrantly processed *miR-127* pre-miRNA in muscle. These would then not be exported from nucleus for further processing into mature miRNAs by the DICER complex. The net effect would be to result in increased levels of RTL1 protein levels (*Ito et al., 2015*). In contrast, loss of SUN1 results in the more efficient processing of the *miR-127* pri-miRNA to a mature functioning miRNA.

In the WT and $Sun1^{-/-}$ grafts, the expression of *Rtl1*, as well as muscle regeneration markers including neonatal- and embryonic myosin heavy chains, and Pax7 were analysed at three time points post grafting (*Figure 6A–D*). *Rtl1* transcript levels were significantly higher in WT grafts than in the $Sun1^{-/-}$ grafts at D5. These then decreased as regeneration proceeded that was largely completed at D14 (*Figure 6A*). At D5, the levels of *Rtl1* correlated with increased expression levels of regeneration markers indicates regeneration was evident in these grafts (*Schiaffino et al., 2015*;

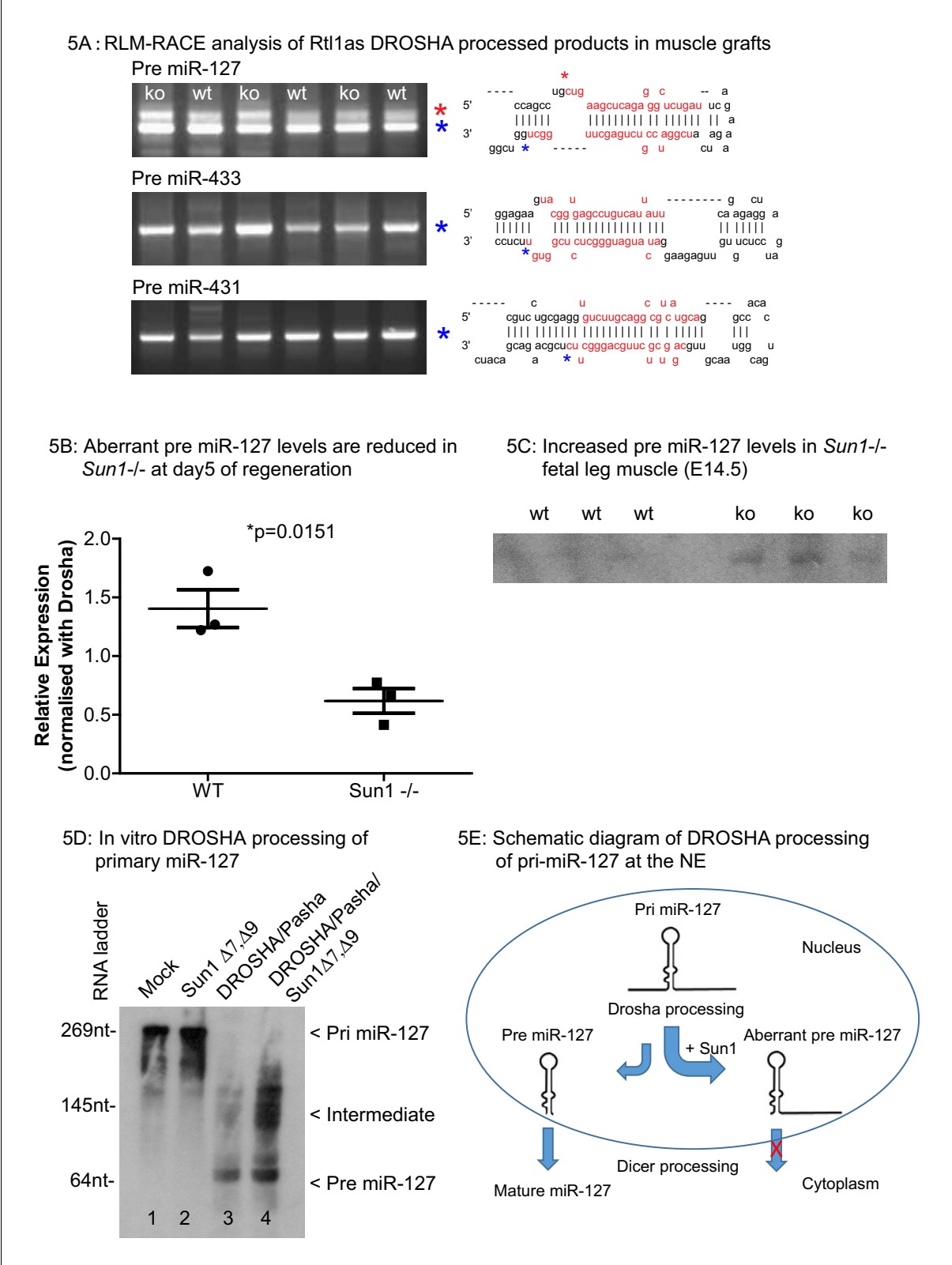

**Figure 5.** RLM-RACE analysis of Rtl1as DROSHA processed products in muscle grafts. (**A**) RLM-RACE was performed on muscle graft samples from D5 of regeneration to identify Drosha mediated cleavage of *Rtl1as* RNA. PCR bands were cloned and sequenced to map Drosha cleavage sites. Sequences of pre-miRNAs in the hairpin configuration and the mature miRNA sequences are highlighted in red, mapped Drosha cleavage sites are marked by asterisks. (**B**) Quantitation from D5 grafts of WT and *Sun1*[−/−] muscle grafts showing aberrant pre-miR-127 levels are reduced in the Sun1[−/−] muscle. (**C**)
*Figure 5 continued on next page*

Figure 5 continued

Total RNA from WT and *Sun1*⁻/⁻ muscle (5 ug) were resolved with 6% TBE-urea gel for detection of pre-miR-127. (D) Drosha, Pasha and SUN1 Δ7, Δ9 proteins were immunoprecipitated from HEK293 cells and incubated with primary miR-127 RNA in vitro. Background non-specific breakdown of primary miR-127 RNA by contaminating RNase from the HEK293 lysate (lanes 1 and 2). Drosha/Pasha cleaved the pri miR-127 to pre miR-127 (lane 3). In the presence of SUN1, Drosha cleaved pri miR-127 into intermediate miR-127 RNA and pre miR-127 (lane 4). (E) Schematic of pri miR-127 and Drosha processing into pre miR-127; the expected product of microprocessor cleavage in nucleus. Pre miR-127 is then exported out of nucleus into cytoplasm for further processing into mature miRNA by Dicer. Sun1 recruits Drosha to the NE and this might disrupts Drosha activity resulting in aberrant processing of pri-miR-127. Aberrant pre-miR-127 might not be exported out in the nucleus resulting in a reduction of mature functional miR-127 RNA.

DOI: https://doi.org/10.7554/eLife.49485.010

The following figure supplement is available for figure 5:

Figure supplement 1. *Rtl1* expression levels were quantified by RT-qPCR in *miR-127* Δ and WT hind-limb muscle from D24 mice.

DOI: https://doi.org/10.7554/eLife.49485.011

*Seale et al., 2000*). We also analysed the expression of 2 other genes in the same gene *Dlk1-Dio3* cluster as *Rtl1*, specifically *Dlk11* and *Meg3* (*Hagan et al., 2009*) in these grafts as *Dkl11* has also been implicated in regulating muscle regeneration (*Figure 6—figure supplement 1*). Although their expression increased in the muscle grafts as they regenerated, their levels did not differ between *Sun1*⁻/⁻ and WT grafts at the three time points sampled.

## *Rtl1* and *Rtl1as* are expressed in regenerating *mdx* muscle and in biopsy samples from patients with EDMD and AD-EDMD

We then determined whether our findings from the regenerating muscle grafts were of relevance to the most prevalent form of muscular dystrophy Duchenne's (DMD). The *mdx* mouse is a model for human DMD, and has ongoing endogenous necrosis of myofibers with associated inflammation, fibrosis and muscle regeneration (*Partridge, 2013*; *Radley et al., 2008*). In the hind-limb muscles of young adult (D24) *mdx* mice, *Rtl1*, *Rtl1as* and its encoded miR-127 were significantly increased in *mdx* as compared to WT littermates (*Figure 6E*). Anti-Rtl1 staining revealed that RTL1 protein was present in the cytoplasm of the regenerated *mdx* myofibers which were the smaller myofibers with central nuclei. Cytoplasmic Rtl1 staining was undetected in the larger *mdx* myofibers with peripheral nuclei (*Figure 6F*). These results indicate that the induction of *Rtl1* and *Rtl1as* expression in newly synthesized myofibers may be a general feature of muscle regeneration.

## Rtl1 is required for efficient muscle regeneration

Since *Rtl1* expression increases in newly formed myofibers, we examined whether loss of *Rtl1* would affect regeneration. We determined the requirement for *Rtl1* during regeneration by autografting EDL muscle from *Rtl1Δ* mice (*Figure 7*) that is in muscle from mice carrying a paternally inherited *Rtl1* deletion (*Ito et al., 2015*). In the *Rtl1Δ* grafts, regeneration was delayed to at least D9 post grafting. In the majority of *Rtl1Δ* grafts, where there was poor clearance of necrotic muscle, fewer regenerated *Rtl1Δ* myofibers were observed compared to WT muscle myofibers. The central graft region was occupied by non-muscle interstitial cell types, indicating defective muscle regeneration in the absence of *Rtl1*. Expression of RTL1 protein was evident in the WT regenerating myofibers identified by peripheral laminin 2- expression and central nuclei localization (*Figure 7C*).

In AD-EDMD muscular dystrophy patients, *miR-127* expression is increased (*Sylvius et al., 2011*). We examined *miR-127* levels in muscle biopsies from six laminopathy patients and four control samples. In the *LMNA* mutants, *miR-127* levels were consistently increased in all six patients relative to the controls (*Figure 7—figure supplement 1*). However, no reliable trend was observed in the expression levels of *RTL1* among the patient samples (data not shown), probably due to variations in the numbers of dystrophic muscle fibers between the individual biopsy samples. The histopathologic description of these muscle biopsies and their origins is summarized in *supplementary file 2*.

## Discussion

Interest in the nuclear lamina and NE has significantly increased over the past few years, primarily because numerous congenital diseases (the lamin or nuclear envelopathies) have been identified which are caused by mutations in many of the NE and lamina associated proteins (*Worman et al.,*

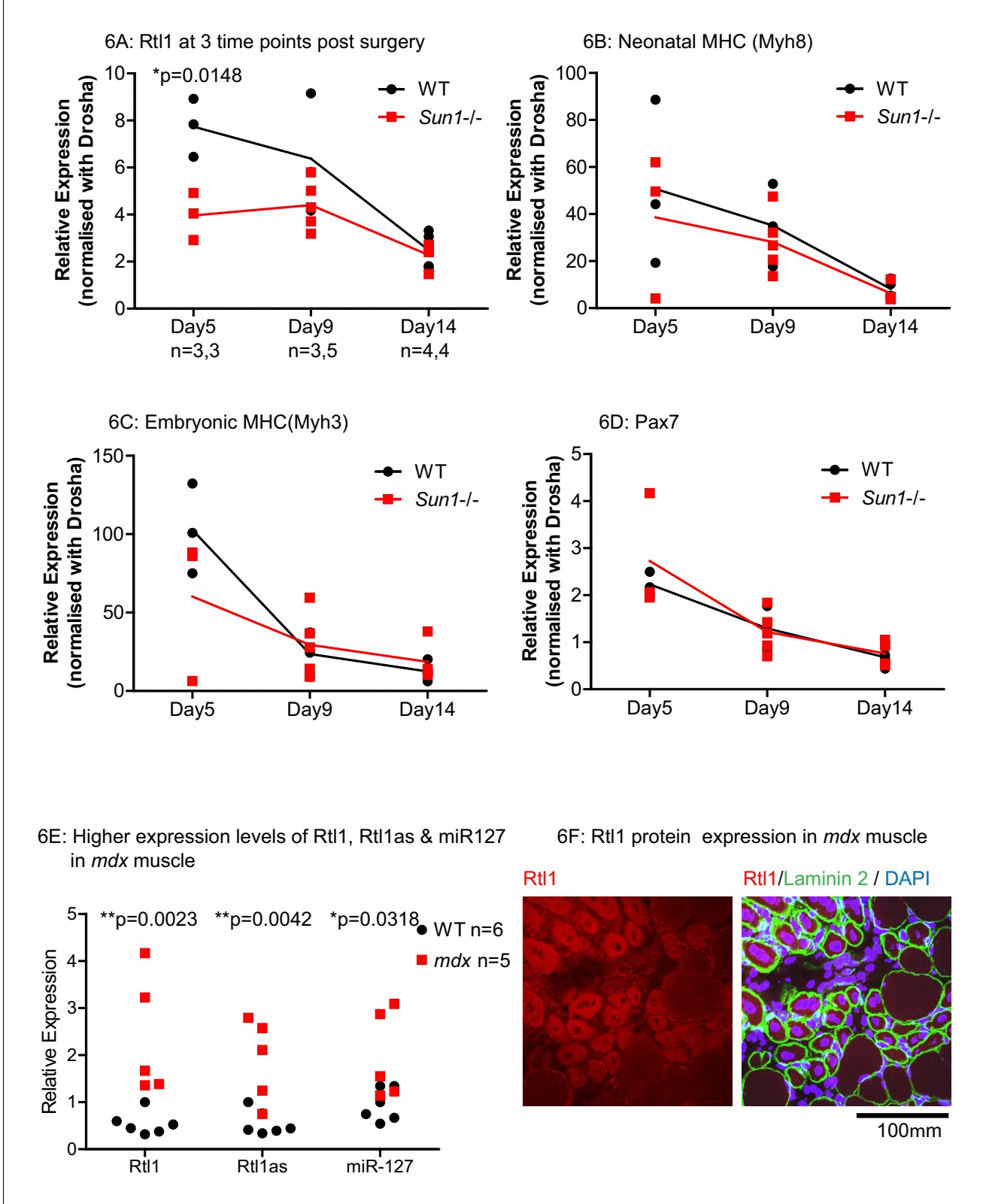

**Figure 6.** Effects of Sun1 loss on Rtl1 and other muscle specific gene expression levels during regeneration. (**A**) *Rtl1* expression increases in regenerating WT myofibers but at lower levels in *Sun1⁻ᐟ⁻* grafts. Both the grafted muscle (left hind-limb) and control muscle (right hind-limb) were harvested at three time-points (Days 5, 9 and 14 post-grafting) for qRT-PCR. The relative expression levels in the muscle grafts over control muscle in each sample with (n) representing sample size is presented at each time-point. WT grafts showed significantly higher *Rtl1* expression compared to

*Figure 6 continued on next page*

*Figure 6 continued*

*Sun1*<sup>−/−</sup> grafts at D5. (**B–D**) High expression levels of *Mhy8, Mhy3* and *Pax7* were noted in these grafts at D5 indicating effective regeneration. (**E**) *Rtl1, Rtl1as* and *miR-127* showed significantly increased levels in the hind-limb muscle of *mdx* mice compared to WT littermates at D24 (postnatal). Statistical analysis with unpaired t test; two-tailed p values *p<0.05 and **p<0.005. (**F**) Confocal image analysis with the anti-Rtl1 antibody revealed RTL1 protein localizes to the cytoplasm of regenerating *mdx* myofibers that have a centrally located nucleus (left panel). Fiber location was determined by co-staining with an anti laminin2 antibody and DAPI (right panel).

DOI: https://doi.org/10.7554/eLife.49485.012

The following figure supplement is available for figure 6:

**Figure supplement 1.** The *D*lk1 and *Meg3* imprinted genes in autografts of WT and *Sun1*<sup>−/−</sup> mice were quantified by RT-qPCR and did not show any significant differences.

DOI: https://doi.org/10.7554/eLife.49485.013

---

*2010*). The majority of these diseases affect muscle (skeletal, vascular and cardiac), with most being caused by mutations in the *LMNA* gene (*Burke and Stewart, 2006*). In addition, mutations/variations in other NE proteins such as *Emerin, Lap2* and the LINC complex protein Nesprin1 (*SYNE1*) may also result in muscle disease (*Burke and Stewart, 2014*). However, to date no mutations in *SUN1* have been linked to any disease, although some variants have been identified which are associated with muscular dystrophy (*Meinke et al., 2014*). *Sun1*<sup>−/−</sup> mice, apart from being infertile and deaf show overtly normal postnatal growth and longevity, with no cardiac or skeletal muscle issues being reported (*Chen et al., 2012*). Intriguingly, loss of some NE proteins such as SUN1 or LAP2α can significantly ammeliorate muscle degeneration caused by *Lmna* mutations, revealing a functional integration of these different NE proteins (*Chen et al., 2012*; *Cohen et al., 2013*).

The primary function of the LINC complexes is to couple the nucleus and nuclear envelope to the cytoskeleton. This coupling is important for nuclear positioning, cytoskeletal organization, cell polarization, and cell migration (*Lee and Burke, 2017*). It also appears that mechanical forces can be transmitted from the extracellular matrix via cell adhesion complexes and the cytoskeleton to the nucleus, resulting in force induced changes to nuclear shape, structure, gene expression, and chromatin composition. In finding that the LINC complex affects these parameters, it has been proposed that the nucleus/NE/LINC complex may function as mechanosensor, in which physical stimuli are converted to changes in gene expression. However how the LINC complex may convert mechanical forces into changes in gene expression is still poorly understood (*Alam et al., 2016*; *Kirby and Lammerding, 2018*).

Here we report that loss of SUN1 delays adult skeletal muscle regeneration, and results in smaller myofiber diameters compared to WT muscle regeneration. Our transplantation experiments suggested that SUN1 isoforms have multiple and possibly distinct functions in muscle regeneration. Some specific SUN1 isoforms may have a role in the inflammatory clearance of necrotic myofibers, a key step in regeneration, as necrotic fibres were not as efficiently cleared in *Sun1*<sup>−/−</sup> grafts compared to WT grafts. The SUN1Δ7 isoform which is specifically enriched in differentiating myoblasts may intrinsically regulate myotube growth in vivo as indicated by the smaller cross sectional area of *Sun1*<sup>−/−</sup> myofibers. To determine how SUN1 may influence muscle regeneration we searched for potential myoblast/myotube nucleoplasmic interaction partners with the predominant muscle isoform Δ7 of SUN1 as bait. Of the ~20 candidates that potentially interact with this SUN1 isoform (*Supplementary file 1*), we focused on the key component of the Microprocessor complex, the RNase III ribonuclease Drosha that initiates microRNA processing in the nucleus.

We find that the recruitment of Drosha to the nuclear envelope is specifically dependent on the SUN1Δ7 isoform in muscle and in transfected fibroblasts (*Figure 3*). With the loss of SUN1 this reduces Drosha recruitment to the NE resulting in the expression levels of at least nine miRNAs being changed (*Figure 4B*). Clearly the issue is how does recruitment by the SUN1Δ7 isoform of Drosha to the NE, specifically alter the expression of a subset of miRNAs, whereas other miRNAs implicated in regulating myoblast proliferation, differentiation and survival (*Chen et al., 2006*; *Cheung et al., 2012*; *Hirai et al., 2010*) were apparently not changed according to our microarray analysis (data not shown).

Within the past few years it has become apparent that the production and levels of mature miRNA's produced by the Microprocessor complex is not a linear process primarily driven by the rate of transcription. The kinetics of processing can vary between different pri-miRNAs, even within the

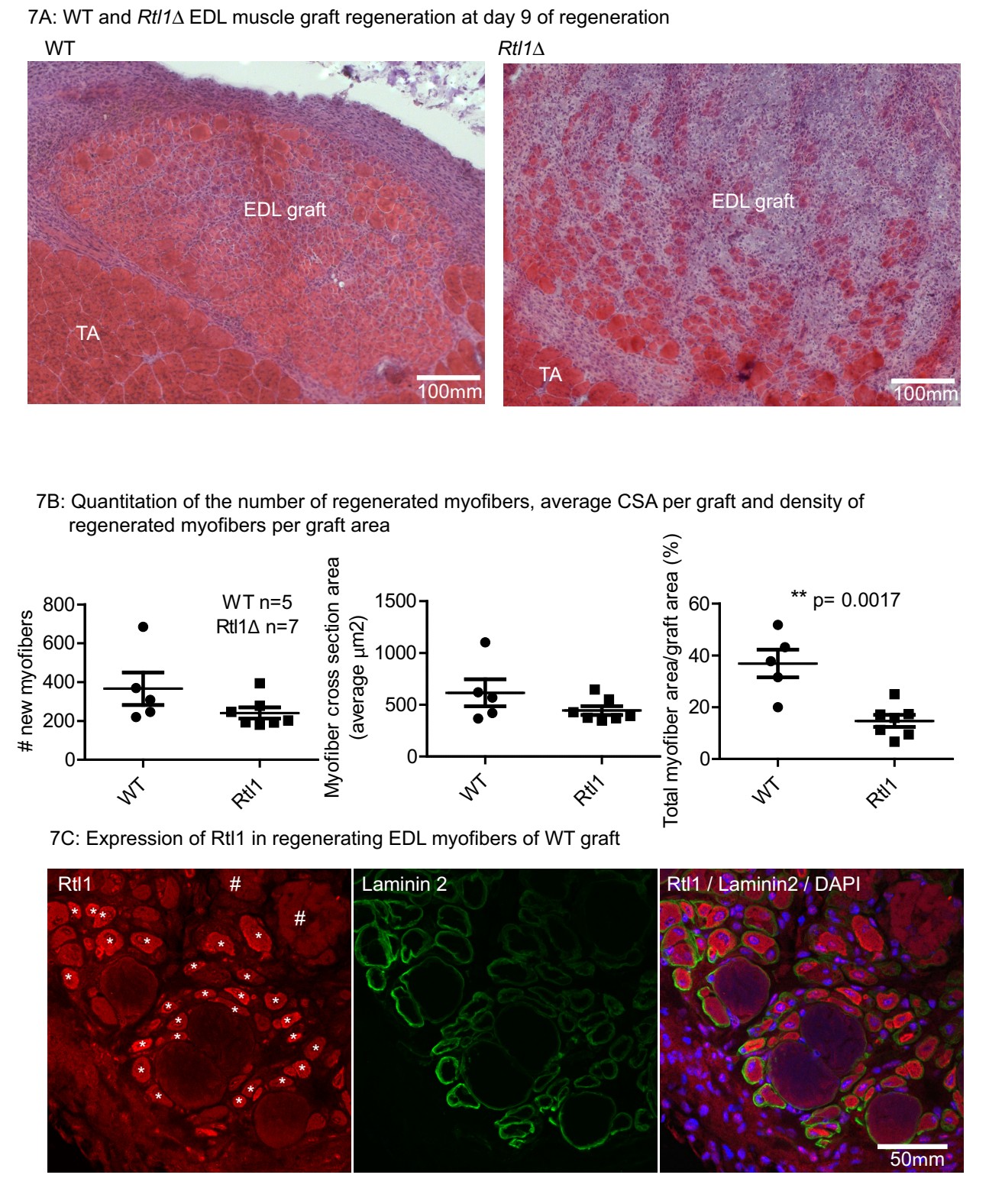

7A: WT and *Rtl1Δ* EDL muscle graft regeneration at day 9 of regeneration

7B: Quantitation of the number of regenerated myofibers, average CSA per graft and density of regenerated myofibers per graft area

7C: Expression of Rtl1 in regenerating EDL myofibers of WT graft

**Figure 7.** Loss of Rtl1 impairs muscle regeneration. (A) H and E images of *Rtl1* Δ and WT EDL autografts at D9 post grafting. (B) The *Rtl1* Δ EDL grafts have fewer regenerated myofibers (p=0.0017) and the central region of the graft is occupied by non-muscle cells, while myogenesis was almost completed in the WT graft. (C) RTL1 protein expression was present in regenerating myofibers (*) as shown by co-staining of anti-RTL1, anti-laminin2 antibody and DAPI. Necrotic muscles are marked with # (C left panel).

*Figure 7 continued on next page*

*Figure 7 continued*

DOI: https://doi.org/10.7554/eLife.49485.014

The following figure supplement is available for figure 7:

**Figure supplement 1.** *miR-127–3* p levels were quantified in healthy human biopsy samples and patients diagnosed with different *LMNA* mutations.
DOI: https://doi.org/10.7554/eLife.49485.015

same cluster/polycistronic transcript, with this variation being due to specific sequences upstream of the miRNA and/or within the hairpin loop. In addition, extrinsic cofactors for example DEAD box helicases such as DDX5 and DDX17 promote processing of individual pri-miRNAs in specific cell types or developmental stages (*Siomi and Siomi, 2010*; *Mori et al., 2014*). Other cofactors suppress cleavage of individual pri-miRNAs, including NF90- NF45 (*Sakamoto et al., 2009*), Lin28B (*Piskounova et al., 2011*), QKI-5 (*Wang et al., 2013*), and MeCP2 (*Cheng et al., 2014*). To these microprocessor co-factors, SUN1, and in particular the muscle enriched Δ7 isoform may now be added as a regulator of miRNA synthesis. In particular, we find that in the presence of SUN1 processing of the *Rtl1as* cluster varied between the different miRNAs encoded in the transcript. We focused on the *Rtl1as* cluster of miRNAs, in particular pri-miR-127, as previously these have been implicated in regulating muscle hypertrophy. The processing of pri-miRNA into pre-miRNA proceeds by the coordinated 5' strand and 3' strand cutting by Drosha RIIIDb and RIIIDa domains respectively (*Han et al., 2004*). RLM-RACE PCR identified the uncoordinated cutting of 5' and 3' strands of pri-miR-127, with the 3' strand being left intact in the presence of SUN1. SUN1 binding to Drosha/ microprocessor may interfere with Drosha RIIIDa domain in a subtle manner as we did not detect any processing abnormalities with the other two pri-miR-433 and pri-miR-431, that are encoded in the same cluster.

This finding expands SUN1's role in controlling the synthesis of small regulatory RNAs, as a previous report indicated a perhaps, indirect, dependency for Piwi interacting RNA (piRNAs) synthesis, another class of small regulatory RNAs found in the germline (*Chi et al., 2009*).

Among the *Rtl1as* encoded miRNAs, the biogenesis of miR-127 was the most efficient compared to the other three miRNAs as shown in the Q-PCR of these miRNAs in the myotubes (*Figure 4C*). This is supported with the in vitro Drosha processing assay where intermediate miR-127 RNAs accumulated in the presence of SUN1 (*Figure 5C*). (*Slezak-Prochazka et al., 2010*; *Choudhury et al., 2013*; *Finnegan and Pasquinelli, 2013*; *Ratnadiwakara et al., 2018*).

We postulate the miRNA profile in in vitro formed myotubes would be different from that of the muscle grafts given the former are pure myotube cultures, while the latter are a complex tissue where muscle fibres are combined with other cell types. Nevertheless, we established the processing of pri-miR-127 (and hence Drosha activity) was sub-optimal in the muscle grafts, and therefore inferred the SUN1-Drosha interaction is physiologically relevant during muscle regeneration. Future studies of pri-miRNAs and their processing into mature miRNAs (of those miRNA listed in *Figure 4B*) should reveal SUN1's mechanistic control of Drosha activity.

*Rtl1as* encoded miRNAs inhibit the post-transcriptional processing of the paternally expressed *Rtl1*, which results in a decrease in *Rtl1* transcripts and consequently RTL1 protein levels (*Davis et al., 2005*; *Xu et al., 2015*; *Ito et al., 2015*). Under normal circumstances SUN1 Δ7,Δ9 inhibition of Drosha processing activity would specifically lead to reduced levels of the four muscle specific *Rtl1as* encoded miRNAs, resulting in the production of appropriate levels of RTL1 protein. With increased *Rtl1as* miRNA levels, *Rtl1* transcript levels were reduced leading to lower RTL1 protein levels (*Ito et al., 2015*). Such a miRNA dependent regulatory mechanism would be an important post-transcriptional control mechanism, in establishing the appropriate levels of RTL1 protein expression for effective muscle regeneration. Excessive RTL1 results in muscle hypertrophy as in adult *callipyge* sheep, in transgenic mice over expressing *Rtl1* and in mice lacking myostatin, a significant inhibitor of muscle growth (*Hitachi and Tsuchida, 2017*). In contrast, loss of *Rtl1*, as we here demonstrate, leads to impaired muscle regeneration.

Together these findings identify RTL1 as being important for muscle growth and regeneration in both mice and sheep with the RTL1 expression levels (translation) being significantly regulated by miRNAs, whose levels, in turn, are determined by interaction with SUN1Δ7 in regulating microprocessor activity. This regulation may be conserved between human and mouse since SUN1 proteins

from both species can bind to Drosha and we showed sequence conservation for the SUN1 Δ7 isoform. RTL1 protein was also detected in the regenerating myofibers of *mdx* muscle, as were transcripts in muscle biopsies from patients diagnosed with Emery-Dreifuss Muscular Dystrophy (AD-EDMD) caused by *LMNA* mutations, indicating that RTL1 expression may be a general requirement of muscle regeneration. How RTL1 protein functions in regulating muscle regeneration is unclear, as other, though conflicting studies, have suggested RTL1 is localized to the nucleus (*Byrne et al., 2010*), whereas another study suggested localization to the cell membrane where it may function as a protease (*Riordan et al., 2013*). In contrast, our immunostaining of RTL1 revealed that it may primarily localize in the myofiber cytoplasm (*Figures 6F* and *7D*), with our co-immunoprecipitation studies indicating a possible interaction between RTL1 and ER protein (unpublished results).

Here we have identified a mechanism by which the LINC complex may mediate gene expression by fine tuning, through the microprocessor complex, the levels of RTL1 expression during muscle regeneration. Whether mechanical forces can directly affect the SUN1 Δ7,Δ9-Drosha interaction and hence the expression of RTL1 remains to be determined. As Lamin A protein also interacts with the nucleoplasmic domain of SUN1, its role in regulating SUN1 Δ7,Δ9-Drosha interactions and whether these are affected by *LMNA* mutations also remains to be established as *LMNA* mutations can increase the levels of SUN1 (*Chen et al., 2012*). If these potential effects do affect miRNA processing this in turn would provide insight into how specific *LMNA* mutations may affect the molecular pathology of tissue specific diseases, in this case muscle.

Besides Drosha, other protein candidates that bind SUN1Δ7,Δ9 (Supplement. file 1) were identified and which have yet to be verified for their role in muscle regeneration can be characterised. Muscle regeneration is a complex process and here we have identified some of the distinct roles of the SUN1 isoforms in muscle regeneration. We would anticipate some of these other identified interaction candidates and SUN1 isoforms to extend SUN1s functions.

## Materials and methods

**Key resources table**

| Reagent type (species) or resource | Designation | Source or reference | Identifiers | Additional information |
|---|---|---|---|---|
| Strain, strain background (*M. musculus*, males and females) | SUN1+/- | PMID: 19211677 | RRID: MGI:3838371 | mixed strain, (C57BL/6J × 129/J); SUN1+ /- males and females for breeding |
| Strain, strain background (*M. musculus*, males and females) | miR-127 Δ | PMID: 26138477 | RRID: MGI:5789800 | C57BL/6J; miR-127 Δ females mated with WT C57BL/6 males for breeding |
| Strain, strain background (*M. musculus*, males) | Rtl1Δ | PMID: 26138477 | RRID: MGI:5789800 | mixed strain, (C57BL/6J × 129/J); miR-127 Δ males mated with WT 129Sv females to get Rtl1Δ pups |
| Strain, strain background (*M. musculus*, males) | mdx | Jackson Laboratory | RRID: IMSR_JAX:001801 | mixed strain, (C57BL/6) |
| Biological sample (*M. musculus*) | Primary myoblast cell (Wildtype and SUN1 -/-) | This paper | | Derived from WT and SUN1 - /- littermates, males |
| Cell line (*H. sapiens*) | Hek-293 | ATCC | CRL-1573TM | |
| Cell line (*M. musculus*) | NIH/3T3 | ATCC | CRL-1658TM | |
| Cell line (*M. musculus*) | C2C12 | ATCC | CRL-1772TM | |

*Continued on next page*

*Continued*

| Reagent type (species) or resource | Designation | Source or reference | Identifiers | Additional information |
|---|---|---|---|---|
| Transfected construct (*M. musculus*) | pCMV6-FLAG-SUN1 Δ7 Δ9 | This paper | | Mammalian expression construct |
| Transfected construct (*M. musculus*) | pCMV6-MYC-SUN1 Δ7 Δ9 | This paper | | Mammalian expression construct |
| Transfected construct (*H. sapiens*) | pCMV6-SUN1 Δ6 -MYC/FLAG | Origene | RC226167 | Mammalian expression construct |
| Transfected construct (*H. sapiens*) | pCMV6-Drosha-GFP | This paper | | Mammalian expression construct |
| Transfected construct (*H. sapiens*) | pFLAG/HA-DGCR8 (Pasha) | PMID: 15589161 | RRID: Addgene_10921 | |
| Recombinant DNA reagent | pSPT18-primary miR127 | This paper | | In vitro transcription of pri-miR127 for Drosha cleavage assay |
| Recombinant DNA reagent | pSPT18-pre- miR127 | This paper | | In vitro transcription, with DIG-labelling of probe |
| Antibody | anti-Drosha (rabbit polyclonal) | Abcam | Cat# ab12286 | IF(1:100), WB (1:500) |
| Antibody | anti-Lamin B (goat polyclonal) | Santa Cruz Biotechnology | Cat# sc-6217 | IF(1:100) |
| Antibody | anti-Lamin A/C (rabbit polyclonal) | Cell Signaling Technology | Cat# 2032 | WB(1:500) |
| Antibody | anti-MYC (rabbit monoclonal) | Cell Signaling Technology | Cat# 2278 | WB(1:2000) |
| Antibody | anti-FLAG (rabbit monoclonal) | Cell Signaling Technology | Cat# 14793 | IF(1:100), WB(1:2000) |
| Antibody | anti-laminin 2 (rat monoclonal) | Enzo life sciences | 4H8-2 | IF(1:200) |
| Antibody | anti-SUN1 12F10 (mouse monoclonal) | This paper | RRID: AB_2813863 | this antibody recognises all mouse SUN1 isoforms |
| Antibody | anti-SUN1 Δ7 (rabbit polyclonal) | PMID: 26417726 | | Raised with peptide RDRTLKPPHLGHC by YenZym Antibodies |
| Antibody | anti-SUN1 Δ7–9 (rabbit polyclonal) | This paper | | Raised with peptide CGGDRTLKPRDLLVQ by YenZym Antibodies |
| Antibody | anti-SUN1 (rabbit polyclonal) | PMID: 19211677 | | this antibody recognises all mouse SUN domain |
| Antibody | anti-Rtl1 (rabbit polyclonal) | This paper | RRID: AB_2813865 | antibody produced by YenZym Antibodies |
| Antibody | anti-GFP 14F5 (mouse monoclonal supernatant) | This paper | provided by Brian Burke, A*STAR | |

## *Sun1*⁻/⁻ and *miR-127*⁻/⁻ breeding and muscle grafting

Mice were maintained at the A*STAR Biological Resource Centre facility in accordance with the guidelines of the IACUC committee. Experimental procedures were performed under the protocol number IUCAC #181326. *SUN1*⁺/⁻ mice were interbred to obtain *SUN1*⁻/⁻. Heterozygous *mdx* females were interbred with wildtype males to obtain WT and *mdx* male pups. The *miR-127* Δ embryos were from Dr. M. Ito. C57BL/6J females carrying the *miR-127* Δ allele were used to maintain the allele. Males carrying a maternally inherited *miR-127* Δ allele were crossed with WT 129Sv

females to obtain *Rtl1* Δ null pups (*Ito et al., 2015*). Only male *Sun1*$^{-/-}$, *Rtl1*Δ and WT littermates were used for muscle grafting. All surgical procedures were performed under avertin anesthesia, with every effort being made to minimize suffering.

Whole muscle autografting and reciprocal grafting were performed to assess the regenerative capacity of myofibers in 2–3 month old *SUN1*$^{-/-}$ and *Rtl1* Δ mice. The Extensor Digitorum Longus (EDL) muscle, with both tendons, was excised and transplanted onto the Tibialis Anterior (TA) muscle. The EDL tendons were sutured to the TA, the skin closed and the wound left to heal (*Roberts and McGeachie, 1992*) (*Shavlakadze et al., 2010*). At day nine post-surgery, mice were euthanized. The TA and grafted EDL were excised, mounted in tragacanth gum (Sigma-Aldrich) on cork pieces and snap frozen with isopentane (BDH-AnalaR) for cryosection. Muscle sections at 7 μm thickness were mounted for histological and immunofluorescence analyses. Grafts were harvested at 5, 9 and 14 days post-surgery for gene expression analysis.

## Immunofluorescence, histology and microscopy

Muscle sections were stained with H and E, and imaged using a Zeiss AxioImagerZ1 microscope and ZEN2 software. Muscle graft analysis and myofiber quantification was performed with Fiji software. For immunofluorescence staining, muscle sections were methanol fixed, permeabilized with 0.1% Triton X-100/PBS, and blocked with BSA/bovine serum. Incubation with primary antibodies was performed at room temperature overnight in a humified chamber. The sections were washed and incubated with secondary Alexa Fluor antibodies and DAPI staining. Imaging was performed using a Zeiss LSM 510 scanning confocal system. Images were processed with LSM image browser software.

## Primary myoblast cultures and preparation of protein lysates

Myoblasts from hindleg muscle were isolated as described (*Grohmann et al., 2005*). Myoblasts were cultured on 0.1% gelatine coated coverslips, and differentiated into myotubes with low serum medium, fixed with ice cold methanol for immuno-staining. Protein lysates were prepared by hypotonic lysis of cells, and centrifuge at 2,000 rpm to pellet the nuclear fraction. The nuclear pellet was resuspended in 1% Triton X-100 lysis buffer with freshly added Roche cOmplete Protease Inhibitor Cocktail to extract nuclear proteins. C2C12 nuclear fraction lysate was used for SUN1 immunoprecipitation experiments.

## Gene expression studies

For total RNA preparation, mouse tissues and cells were homogenised in Invitrogen TRIzol Reagent and chloroform. After centrifugation, the supernatant was passed through Qiagen RNeasy Mini spin columns to yield total RNA. For human samples, total RNA was prepared with Exiqon miRCURY RNA isolation kit. For miRNA RT-qPCR, 200 ng of total RNA was reverse transcribed (RT) with Exiqon miRCURY LNA universal RT kit. qPCR were performed with Fast SYBR Green Master Mix and Exiqon LNA primers. For mouse samples, cDNA was synthesized from 1 μg of total RNA with Multi-Scribe reverse transcriptase and random hexamers (Invitrogen). Primer sequences are listed in *Supplementary file 3* Table 3. qPCR experiments were performed in triplicates, and relative expression was calculated by the ddCT method. Total RNA from WT and *Sun1*$^{-/-}$ myotubes (one day after differentiation) were processed by Exiqon for miRNA microarray analysis. miRNA validation was performed in triplicate. Statistical analysis by the unpaired t- test with two tailed p values. A P value < 0.05 was considered significant.

## Cloning of Sun1 isoforms, Y2H and transient transfection

Mouse *Sun1* cDNAs encoding the N-terminal nucleoplasmic domain were reverse transcribed with AccuScript RT and Sun1 primer (CTACTG GATGGGCTCTCCGTGGAC) from total RNA, and PCR amplified (primer sequences in *Supplementary file 3* Table 3). Human SUN1 cDNA was PCR amplified from human skeletal muscle and fetal muscle Clontech Marathon-Ready cDNAs. The nucleoplasmic domain of Sun1 Δ 7, Δ nine was used as bait to screen for potential interactors using the ProNet Y2H system (Myriad Genetics, Inc). Different clones encoding Drosha protein domain spanning amino acids 57–674 and 204–727 were screened from uterus/mammary gland- and brain libraries respectively. Sun1 Δ 7, Δ nine and *Drosha* cDNAs were cloned into Origene pCMV6 vectors for HEK293 transient transfection with Lipofectamine 2000. *Pasha* cDNA was from Addgene plasmid

10921 (*Landthaler et al., 2004*). Cells were harvested the next day, lysed with 1% Triton X-100 lysis buffer containing Roche Complete Protease Inhibitor Cocktail and sonicated. Transient transfection of NIH3T3 was performed using Lipofectamine 3000 reagent.

## Drosha cleavage assay

The primary miR-127 sequence was cloned in pSPT18 vector. For each assay, the clone was linearized for in vitro transcription (Promega) to produce the primary (pri-) miR-127 RNA hairpin. Pasha, Drosha and SUN1 proteins were immuno-precipitated from HEK293 cells and incubated with the pri miR-127 RNA hairpin. This procedure was performed within 3 hr to ensure maximal Drosha activity. RNA was resolved with Novex 6% TBE - urea gel, RNA was transblotted to Nylon membrane for overnight hybridisation with DIG-labelled probe. The probe sequence is complementary to the pre miR-127 sequence, and the hybridised probe on Nylon membrane was detected with anti-DIG AP and CDP* (DIG Northern Starter Kit, Roche). The same procedure was performed for the detection of pre-miR-127 in muscle, 5 ug of total RNA of muscle samples was resolved with 6% TBE - urea gel as above.

## Rlm-race

2–3 ug of total RNA from muscle grafts were ligated to 0.25 ug of GeneRacer RNA adaptor (CGAC UGGAGCACGAGGACACUGACAUGGACUGAAGGAGUAGAAA) at 16˚C for 6 hr. The ligated RNA was reverse transcribed with AccuScript RT and primer (GCGGGCCCTGGTGGACTCAGGAGC) to amplify Rtl1as containing miR-127, miR-433 and miR-431. Subsequently two rounds of PCR was performed to enrich for the miRNA sequences; the first PCR performed with the forward primer (GeneRacer 5′ Primer) which anneals to the adaptor sequence and reverse primer anneal to Rtl1as (CCCA TGCCCCTGAAGTCGACTGGA), the second PCR was performed with nested primers (GeneRacer 5′ Nested Primer and Rtl1as primer sequences are listed in *Supplementary file 3* Table 3).

## Antibodies

Murine sequence specific SUN1 and RTL1 antibodies were raised in rabbits and affinity purified by YenZym Antibodies, CA. The immunogen was the RDRTLKPPHLGHC peptide for mouse SUN1Δ7 and His tagged fusion protein for mouse RTL1 aa89-315. Specificity of RTL1 antibodies were confirmed by Western blot with WT and *Rtl1* Δ fetus and placental extracts (*Ito et al., 2015*). The SUN1 Δ seven antibody showed no cross-reactivity with other SUN1 isoforms (*Calvi et al., 2015*). SUN1 antibodies raised against the Sun domain was provided by Dr Ya-Hui Chi (National Health Research Institutes, Taiwan), SUN1 monoclonal antibodies (12.10F) was raised against exon six sequences. GFP monoclonal antibodies was provided by Dr Brian Burke (A*STAR, Singapore). Antibodies to Drosha (Abcam ab12286); Lamin B1 (Santa Cruz Biotechnology); Lamin AC and MYC-tag (Cell signalling technology); M2 FLAG and laminin2 (ENZO); Alexa Fluor-conjugated (Invitrogen) and HRP-conjugated (Dako) secondary antibodies.

## Acknowledgements

We are very grateful to Dr Prabha Sampath for advice and extensive comments on the manuscript. This work was supported by the Singapore Agency for Science, Technology and Research (A*STAR) and by the Singapore Biomedical Research Council Translational Clinical Research grant NMRC/ TCR/006-NUHS/2013 to CLS. We also thank the Biological Resource Centre, A*STAR Singapore that re-derived the miR-127 null mice from embryos provided by Dr Mitsuteru Ito.

## Additional information

### Funding

| Funder | Grant reference number | Author |
| --- | --- | --- |
| National Medical Research Council | NMRC/TCR/006-NUHS/2013 | Colin L Stewart |
| Agency for Science, Technology and Research | | Colin L Stewart |

The funders had no role in study design, data collection and interpretation, or the decision to submit the work for publication.

## Author contributions

Tsui Han Loo, Conceptualization, Data curation, Formal analysis, Investigation, Methodology, Writing—review and editing; Xiaoqian Ye, Data curation, Formal analysis, Methodology, Performed many of the molecular analyses; Ruth Jinfen Chai, Resources, Data curation, Methodology, Provided expertise and guidance in the muscle transplant experiments and analysis; Mitsuteru Ito, Resources, Formal analysis, Investigation, Methodology; Gisèle Bonne, Resources, Data curation, Investigation, Methodology; Anne C Ferguson-Smith, Conceptualization, Investigation, Writing—review and editing; Colin L Stewart, Conceptualization, Supervision, Funding acquisition, Writing—original draft, Project administration, Writing—review and editing

## Author ORCIDs

Tsui Han Loo (iD) https://orcid.org/0000-0002-5489-6469
Colin L Stewart (iD) https://orcid.org/0000-0002-4988-536X

## Ethics

Animal experimentation: Mice were maintained at the A*STAR Biological Resource Centre facility in accordance with the guidelines of the IACUC committee. Experimental procedures were performed under the protocol number IUCAC #181326.

## Decision letter and Author response

Decision letter https://doi.org/10.7554/eLife.49485.022
Author response https://doi.org/10.7554/eLife.49485.023

## Additional files

### Supplementary files

• Source data 1. Raw data for graphs.
DOI: https://doi.org/10.7554/eLife.49485.016

• Supplementary file 1. SUN1Δ7, Δ9 Y2H interaction candidates and libraries screened.
DOI: https://doi.org/10.7554/eLife.49485.017

• Supplementary file 2. Clinical phenotypes and genotypes of *LMNA* mutated patients. LGMD1B: Limb girdle muscular dystrophy type 1B; DCM: Dilated cardiomyopathy; CD: Conduction disease; CD-ARRH: conduction defects with arrhythmias, EDMD; Emery-Dreifuss muscular dystrophy; ICD: implantable cardiac defibrillator, PLD: Partial lipodystrophy.
DOI: https://doi.org/10.7554/eLife.49485.018

• Supplementary file 3. List of primer sequences.
DOI: https://doi.org/10.7554/eLife.49485.019

• Transparent reporting form DOI: https://doi.org/10.7554/eLife.49485.020

### Data availability

All data generated or analysed during this study are included in the manuscript and supporting files. Source data files have been provided for all Figures.

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
