## [Decision Letter]

**Acceptance summary:**

Loo and colleagues present a conceptually novel and exciting study, as it suggests an interesting mechanism by which the nuclear envelope can impinge on regulating gene expression. The study has mechanistic depth as it uncovers multiple layers of interconnected mechanisms mediated by *Sun1* in regulating muscle regeneration.

**Decision letter after peer review:**

Thank you for submitting your article "The LINC complex component SUN1 regulates muscle regeneration by modulating Drosha activity" for consideration by *eLife*. Your article has been reviewed by three peer reviewers, and the evaluation has been overseen by a Reviewing Editor and Kathryn Cheah as the Senior Editor. The reviewers have opted to remain anonymous.

The reviewers have discussed the reviews with one another and the Reviewing Editor has drafted this decision to help you prepare a revised submission.

Summary:

Loo and colleagues present an intriguing paper in which they describe a novel mechanism mediated by *Sun1* in regulating skeletal muscle regeneration. They show that this regulation is dependent on the ability of specific *Sun1* isoforms to bind to Drosha and Pasha, which are components of the miRNA processing machinery. They further show that this interaction is necessary to reduce the processing of antisense *Rtl1* into miRNAs that target the production of RTL1 protein, which has an established role in the maintenance of materno-fetal interface but only a suggestive role in muscle. Lastly, they present supportive data in the *mdx* mouse model of muscular dystrophy and in skeletal muscle biopsies from patients with AD-EDMD, although no clear pattern was noted presumably due to the heterogeneity of patient samples.

The concept of the study is novel and implications from the results are exciting as it suggests another mechanism in the repertoire by which the nuclear envelope can impinge on regulating gene expression. The study has mechanistic depth as it uncovers multiple layers of interconnected mechanisms mediated by *Sun1* in regulating muscle regeneration.

Essential revisions:

1) Include Pasha binding in Sun 1 Δ7 and Δ9 in Figure 3D.

2) Test increased nuclear miR-127 pre-miRNA in muscle in the SUN1 KO mice.

3) In vitro silencing of SUN1 protein in primary WT myoblasts or in C2C12 cell line leading to increased processing of *Rtl1*as that reduce RTL1 expression (either mRNA or protein) would help fortify the mechanistic link between SUN1 and RTL1 expression.

4) What is the epistatic relationship between the players identified in this work? If SUN1 is inhibiting/sequestering Drosha, then Drosha over-expression should overcome the defect observed in cells lacking SUN1. Similarly, if insufficient levels of *Rtl1* drives the observed defects in muscle regeneration, then increasing *Rtl1* levels (by over-expressing, or by removing miR-127 – the latter seems possible because there is an existing model) should rescue the *Sun1^-/-^.* Additional insight could come from exploring the epistatic relationship between *Sun1, Rtl1* and miR-127 (perhaps addressed by engrafting given that the authors have mouse models in hand?).

---

## [Author Response]

Essential revisions:

*1) Include Pasha binding in Sun 1* Δ*7 and* Δ*9 in Figure 3D.*

We have now included Pasha binding in Figure 3D where FLAG tagged Pasha was immuno-precipitated and showed co-precipitation with Drosha and mouse SUN1.

2) Test increased nuclear miR-127 pre-miRNA in muscle in the SUN1 KO mice.

We were not quite sure what was meant by this revision. However we now show that there is an increase in pre-miR-127 levels in fetal *Sun1* KO muscle compared to WT fetal muscle samples as presented in Figure 5C.

3) In vitro silencing of SUN1 protein in primary WT myoblasts or in C2C12 cell line leading to increased processing of Rtl1as that reduce RTL1 expression (either mRNA or protein) would help fortify the mechanistic link between SUN1 and RTL1 expression.

This was an excellent suggestion. However in Figure 4—figure supplement 1 we show (by qPCR and Western blot) that only very low levels or close to non-detectable levels of RTL1 protein are expressed in cultured primary myoblasts and C2C12 cells (Figure 4—figure supplement 1). As a reference, fetal muscle shows detectable levels of *Rtl1* protein and transcripts (Figure 4—figure supplement 1). Consequently we felt it would be experimentally difficult to demonstrate that SUN1 knockdown with shRNA would lead to further reductions in *Rtl1* levels as these adult myoblast cells already exhibited low/no *Rtl1* expression.

4) What is the epistatic relationship between the players identified in this work? If SUN1 is inhibiting/sequestering Drosha, then Drosha over-expression should overcome the defect observed in cells lacking SUN1. Similarly, if insufficient levels of Rtl1 drives the observed defects in muscle regeneration, then increasing Rtl1 levels (by over-expressing, or by removing miR-127 – the latter seems possible because there is an existing model) should rescue the Sun1^-/-^. Additional insight could come from exploring the epistatic relationship between Sun1, Rtl1 and miR-127 (perhaps addressed by engrafting given that the authors have mouse models in hand?).

According to our proposed model, overexpressed Drosha should process the *Rtl1*as more efficiently, as we indeed see in *Sun1* null muscle where SUN1 is no longer inhibiting Drosha activity. However Drosha protein levels are tightly regulated by Pasha and Pasha RNA transcripts are degraded by Drosha RNase activity (Han et al., 2009, Cell, 136, 75-84). With this type of cross regulation it may not be feasible to overexpress Drosha.

We would predict transplanting miR-127 null EDL (which would be overexpressing *Rtl1*) into SUN1 null muscle may not rescue muscle regeneration. We showed that transplanted WT EDL muscle into a SUN1 null host and the reciprocal experiment did not rescue muscle regeneration (Figure 2—figure supplement 2A). Our rationale is that there may be at least 2 or more *Sun1* protein isoforms (due to alternative splicing) expressed in the muscle during regeneration, where they play distinct roles in necrotic muscle clearance, inflammatory responses and myogenesis. SUN1Δ7 D9 -Drosha is one isoform we focused on in this manuscript. Future analyses of the other SUN1 binding proteins that we identified by Y2H (Supplementary file 1) may yield further insights into the multiple roles played by SUN1 isoforms/protein interactions in muscle regeneration. Furthermore, the differential microRNA profile (besides the ones encoded by *Rtl1*as) between WT and *Sun1* null would also likely have a role in muscle regeneration as well.